# Apoptotic cells can induce non-autonomous apoptosis through the TNF pathway

Ainhoa Pérez-Garijo, Yaron Fuchs, Hermann Steller*

Strang Laboratory of Apoptosis and Cancer Biology, Howard Hughes Medical Institute, The Rockefeller University, New York, United States

**Abstract** Apoptotic cells can produce signals to instruct cells in their local environment, including ones that stimulate engulfment and proliferation. We identified a novel mode of communication by which apoptotic cells induce additional apoptosis in the same tissue. Strong induction of apoptosis in one compartment of the *Drosophila* wing disc causes apoptosis of cells in the other compartment, indicating that dying cells can release long-range death factors. We identified Eiger, the *Drosophila* tumor necrosis factor (TNF) homolog, as the signal responsible for apoptosis-induced apoptosis (AiA). Eiger is produced in apoptotic cells and, through activation of the c-Jun N-terminal kinase (JNK) pathway, is able to propagate the initial apoptotic stimulus. We also show that during coordinated cell death of hair follicle cells in mice, TNF-α is expressed in apoptotic cells and is required for normal cell death. AiA provides a mechanism to explain cohort behavior of dying cells that is seen both in normal development and under pathological conditions.

*For correspondence: steller@rockefeller.edu

**Competing interests:** The authors declare that no competing interests exist.

## Introduction

Apoptosis is a distinct form of programmed cell death in which cells activate an intrinsic suicide program to self-destruct. This process plays a major role in development and tissue homeostasis, and abnormal regulation of apoptosis is associated with a variety of human diseases (*Fuchs and Steller, 2011*). Dying cells can secrete signals that will stimulate the recruitment of phagocytes (find-me signals) as well as expose signals on their surface to facilitate their engulfment (eat-me signals) (*Lauber et al., 2003*; *Ravichandran, 2003*). However, apoptosis has been traditionally regarded as a silent process that does not affect surrounding tissues. Only more recently has it become clear that apoptotic cells are the source of signals that can have profound effects on their neighbors. Cells that undergo apoptosis in response to stress and injury can secrete mitogenic and morphogenetic signaling proteins to stimulate growth and tissue repair (*Bergmann and Steller, 2010*; *Morata et al., 2011*; *Greco, 2013*). These factors include Wnt, Dpp/Bmps and Hedgehog (Hh) proteins, which all play major roles in the regulation of growth and patterning during development (*Huh et al., 2004*; *Perez-Garijo et al., 2004*; *Ryoo et al., 2004*; *Fan and Bergmann, 2008*). Mitogenic signaling by apoptotic cells has been reported for a diversity of animals, from *Hydra*, to flat worms, *Drosophila* and vertebrates, and it has been implicated in regeneration, wound healing and tumor growth (*Tseng et al., 2007*; *Chera et al., 2009*; *Bergmann and Steller, 2010*; *Li et al., 2010*; *Pellettieri et al., 2010*; *Huang et al., 2011*). This mechanism appears well suited to communicate cellular loss to stem and progenitor cells in the tissue environment to stimulate proliferation and tissue repair. On the other hand, large groups of cells often undergo coordinated death during development and under conditions of severe tissue injury (*Glucksmann, 1951*; *Jacobson et al., 1997*). Classic examples for such group suicide behavior in normal development include the elimination of the tadpod tail during amphibian metamorphosis, and the removal of interdigital membranes during digit individualization in vertebrates. In *Drosophila*, apoptosis plays a crucial role in

**eLife digest** The tissues of developing organisms can be shaped by apoptosis, a form of regulated cell killing. Although this process can occur in individual cells, apoptotic signals may also dictate the 'communal death' of many cells simultaneously. This occurs frequently in animal development: in human fetuses, for example, cells in the hand are directed to die to remove webbing between the fingers.

Apoptosis has been thought to resemble a form of silent suicide by cells, but more recent work suggests that apoptotic cells can also transmit signals. Now, Pérez-Garijo et al. find that these cells can stimulate other cells to die in both fruit flies and mice.

In fruit flies, apoptosis is activated by proteins known as Grim, Hid and Reaper. To explore whether apoptotic cells could communicate with other cells, Pérez-Garijo et al. created 'undead' cells in which one of these proteins was turned on, but other downstream proteins (that are responsible for the cellular execution phase of apoptosis) had been turned off: these cells were undergoing apoptosis, but could not complete the process and die.

Strikingly, undead cells in the posterior (back) region of the wing imaginal disc—the tissue in the larva that gives rise to the wing in the adult fruit fly—could trigger apoptosis in cells in the anterior (front) half. Pérez-Garijo et al. found that the JNK pathway activated apoptosis in anterior cells. In fruit flies, the Eiger protein turns on this pathway; when Eiger was absent from posterior cells in the wing imaginal disc, apoptosis in anterior cells ceased, indicating that Eiger might signal at long range.

Eiger is related to a protein called TNF that has been implicated in cycles of destruction and renewal of hair follicles in mice. Pérez-Garijo et al. found that TNF is produced by apoptotic cells in hair follicles, and that blocking TNF inhibits the death of other cells in the same cohort: this suggests that a common mechanism could regulate the communal death of cells in flies and mammals. These studies therefore shed light on a conserved pathway in the modulation of tissue development.

several morphogenetic events, sculpting tissues and organs, removing a large number of cells in a coordinated manner and inducing cellular reorganization (*Lohmann et al., 2002*; *Link et al., 2007*; *Manjon et al., 2007*; *Suzanne et al., 2010*; *Suzanne and Steller, 2013*). In vertebrates, another example of communal death is the regressive phase of the hair follicle (HF), which undergoes cycles of growth (anagen), degeneration (catagen) and rest (telogen) (*Hardy, 1992*; *Fuchs, 2007*). In catagen, all the cells in the lower portion of the HF are eliminated by apoptosis (*Lindner et al., 1997*; *Botchkareva et al., 2006*). In all these cases, cell death takes place in a very rapid and highly synchronized manner. However, it is not known how this cohort behavior is achieved. Likewise, many pathological states are associated with extensive cell death, which leads to severe damage and can have grave consequences for patients. Examples include alcohol-/drug-induced liver failure, viral infection, cardiac infarction, ischemic stroke and cachexia (*Sharma and Anker, 2002*; *Kang and Izumo, 2003*; *Guicciardi and Gores, 2005*; *Yuan, 2009*). In all these pathologies, apoptosis accounts for widespread cell loss and is thought to contribute to patient mortality (*Thompson, 1995*; *Favaloro et al., 2012*). One possible explanation for all these 'mass suicide' phenomena is that apoptotic cells may release signals to coordinate their 'communal death'.

Here we investigated whether apoptotic cells are able to produce signals that can explain the coordinated behavior of groups of dying cells. We observed that massive induction of apoptosis in the posterior compartment of *Drosophila* wing discs caused non-autonomous apoptosis at a considerable distance in the anterior compartment. Moreover, apoptosis of cells in the anterior compartment requires signaling from apoptotic cells in the posterior compartment, indicating that apoptosis-induced-apoptosis (AiA) is an active phenomenon. We next explored the mechanism underlying AiA and found that apoptotic cells produce Eiger, the TNF homolog in *Drosophila*. Eiger activates the JNK pathway in neighboring cells and induces them to die. Finally, we examined whether AiA also occurs in vertebrates and whether it plays a physiological role for the coordinated death of groups of cells. We found that during the regressive phase (catagen) of the HF, apoptotic cells produce TNF-α. Inhibition of TNF-α disrupts the coordinated death of HF cells in catagen, indicating

that this mechanism plays a physiological role to maintain synchronicity in the HF cycle. Taken together, these observations reveal a novel mechanism to coordinate cohort behavior of dying cells that is seen both in normal development and under pathological conditions.

## Results

### Undead cells promote non-autonomous apoptosis

To reveal novel types of signaling induced by apoptotic cells we made use of 'undead' cells: apoptotic cells that are kept alive by the expression of the baculovirus caspase inhibitor p35 (*Hay et al., 1994*). Under these conditions, cells initiate the apoptotic cascade but cannot execute cell death because the activity of effector caspases is blocked. Although apoptotic cells are normally very rapidly cleared in living tissues, undead cells persist for long times and thereby permit analysis of signaling events associated with the induction of apoptosis. We used the Gal4/UAS system for transgene expression of the pro-apoptotic gene *hid* and the caspase inhibitor *p35* in the posterior compartment of wing imaginal discs (*Brand and Perrimon, 1993*). As expected, we observed ectopic Wg expression and discs with abnormal and in many cases overgrown posterior compartments due to increased cell proliferation (*Figure 1*). Undead cells contain high levels of cleaved caspases and were visualized by staining with activated caspase-3 antibody, which recognizes cleaved effector caspases as well as the activity of the

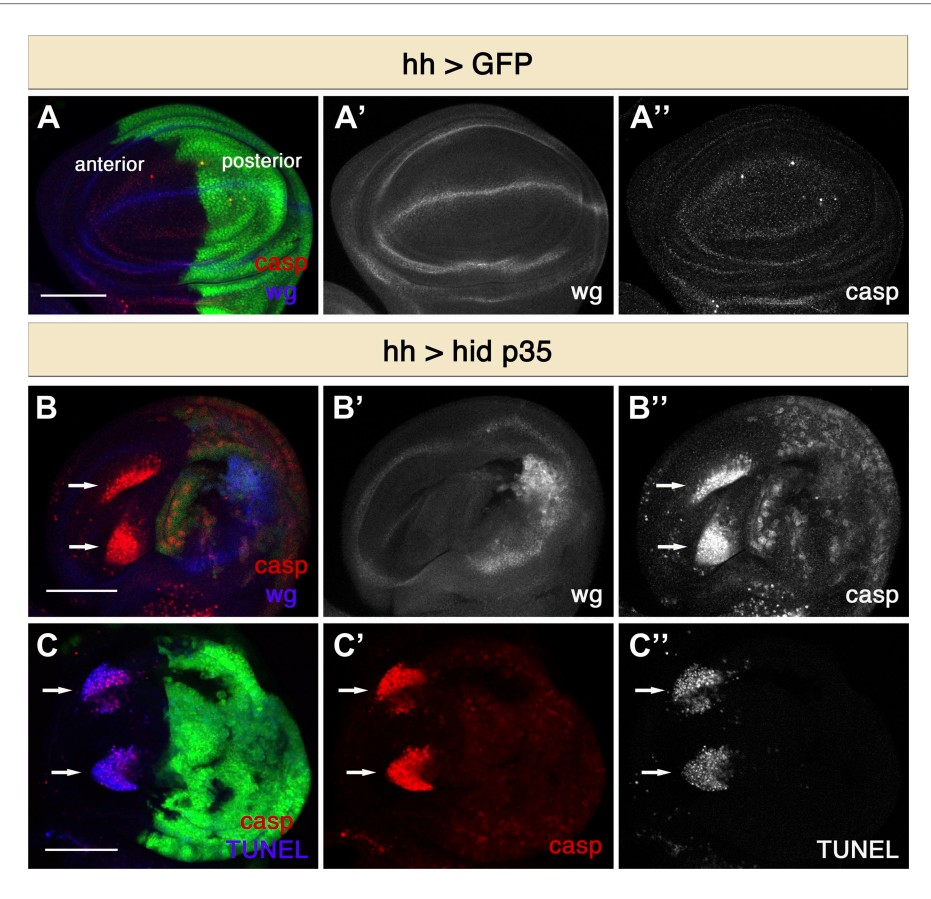

**Figure 1**. Undead cells promote apoptosis in neighboring cells. (**A**) Wing disc of the genotype *hh-Gal4>UAS-GFP* showing wild-type *wg* expression (blue in **A**, white in **A'**) and normal levels of apoptosis, visualized by cleaved-caspase-3 (casp) staining (red in **A**, white in **A''**). (**B**) Wing discs of the genotype *hh-Gal4>UAS-GFP UAS-hid UAS-p35*. Undead cells in the posterior compartment show ectopic *wg* activation (blue in **B**, white in **B'**) and are labeled by diffuse cleaved-caspase-3 staining (red in **B**, white in **B''**). Non-autonomous apoptosis was observed in the anterior compartment (arrows). (**C**) Apoptotic cells in the anterior compartment (arrows) show cleaved-caspase-3 staining (red) and TUNEL staining (blue in **C**, white in **C''**). Scale bars: 100 µm.

initiator caspase Dronc (*Figure 1B*) (*Fan and Bergmann, 2010*). Surprisingly, we also observed large numbers of apoptotic cells in the anterior compartment (*Figure 1B*). Under these conditions, we typically saw two large clusters of dying cells in the wing pouch. It appears that cells in this region of the wing disc are more susceptible to apoptosis, as indicated by the fact that higher rates of cell death within this region were also observed after X-irradiation and *hid* over-expression (*Milan et al., 1997*; *Moon et al., 2005*). Interestingly, caspase-3 staining of apoptotic cells in the anterior compartment differed significantly from that seen in undead cells. Although active caspase-3 immunoreactivity was cytoplasmic and diffuse in undead cells, the staining of cells in the anterior compartment was punctate and intense, indicating that these cells are dying (*Figure 1B*). To confirm this idea, we performed TUNEL labeling (*Figure 1C*). As expected, undead cells in the posterior compartment did not show TUNEL staining, but caspase-3-positive cells in the anterior compartment also displayed distinct TUNEL labeling (*Figure 1C*). These findings indicate that undead cells in the posterior compartment of the *Drosophila* wing disc have the ability to stimulate the induction of apoptosis at a distance in a different compartment.

To examine whether the ability of undead cells to induce non-autonomous apoptosis is more general, we used different paradigms to generate undead cells. Expression of the pro-apoptotic gene *rpr* along with *p35* using the same driver also produced extensive cell death in the anterior compartment (not shown). Furthermore, non-autonomous apoptosis is not restricted to the wing imaginal disc, as we also observed apoptosis in the anterior compartment of other discs, such as the haltere or the leg discs (*Figure 2A,B*). On the other hand, we did not observe apoptosis in the eye-antennal discs, suggesting this is not a general systemic response (*Figure 2C*). However, this phenomenon is not compartment specific. We used the apterous-Gal4 (ap-Gal) driver to express *hid* and *p35* in the dorsal compartment of wing discs. In this case, we observed widespread apoptosis in the ventral compartment (*Figure 2D*). However, the use of weaker drivers (such as Ci-Gal4 and en-Gal4) produced very little non-autonomous apoptosis. This suggests that a strong apoptotic stimulus is required to induce non-autonomous apoptosis.

We next wanted to confirm that the observed anterior apoptosis is in fact non-autonomous in contrast to undead cells that might have migrated from the posterior compartment and escaped the *p35* protection. For this aim, we utilized the Q transgene expression system (*Potter et al., 2010*; *Potter and Luo, 2011*) in combination with the Gal4 system, enabling us to independently control the expression of QF and Gal4 in the posterior and the anterior compartments, respectively (*Figure 3A*). We then expressed *rpr* and *p35* in the posterior compartment under the control of the QF and use Ci-Gal4>UAS-GFP to label the anterior compartment. Undead cells can be visualized in the posterior compartment by staining with cleaved caspase-3 antibody (*Figure 3B*). Importantly, we observed that cells that die in the anterior compartment express the anterior marker Ci-Gal4, demonstrating that they are of anterior origin, and not 'escaping undead cells' (*Figure 3B*). This confirms that the apoptosis taking place in the anterior compartment is indeed non-autonomous apoptosis.

## Non-autonomous apoptosis is induced by signaling from apoptotic cells

*Milan et al. (1997)* reported non-autonomous apoptosis upon ricin overexpression, which they explained as being an indirect consequence of changes in proliferation during compartment size accommodation. Since in our experiments posterior compartments were usually enlarged, we considered the possibility that non-autonomous apoptosis is the consequence of a size compensation mechanism across compartments to maintain the overall size of the disc. To investigate this possibility, we manipulated the size of the posterior compartment by downregulation of dMyc. dMyc is a key regulator of tissue growth, and downregulation of dMyc leads to a reduction of cell and tissue size (*Johnston et al., 1999*). We reasoned that if non-autonomous apoptosis is the consequence of the increased size of the posterior compartment, then this phenomenon should not take place under conditions where the size of the posterior compartment is normal or reduced. Therefore, we generated posterior compartments that express *hid* and *p35* together with dMyc RNAi. Under these conditions, we still observed undead cells, as revealed by cleaved caspase-3 staining, in a compartment with significantly reduced size (*Figure 4A*). Importantly, despite the size reduction, we still saw large numbers of apoptotic cells in the anterior compartment (*Figure 4A*). These observations argue strongly against the idea that size compensation causes non-autonomous cell death in this system.

We also considered the possibility that non-autonomous apoptosis could be a consequence of the production of Dpp and Wg by undead cells, which could distort the morphogen gradient leading to

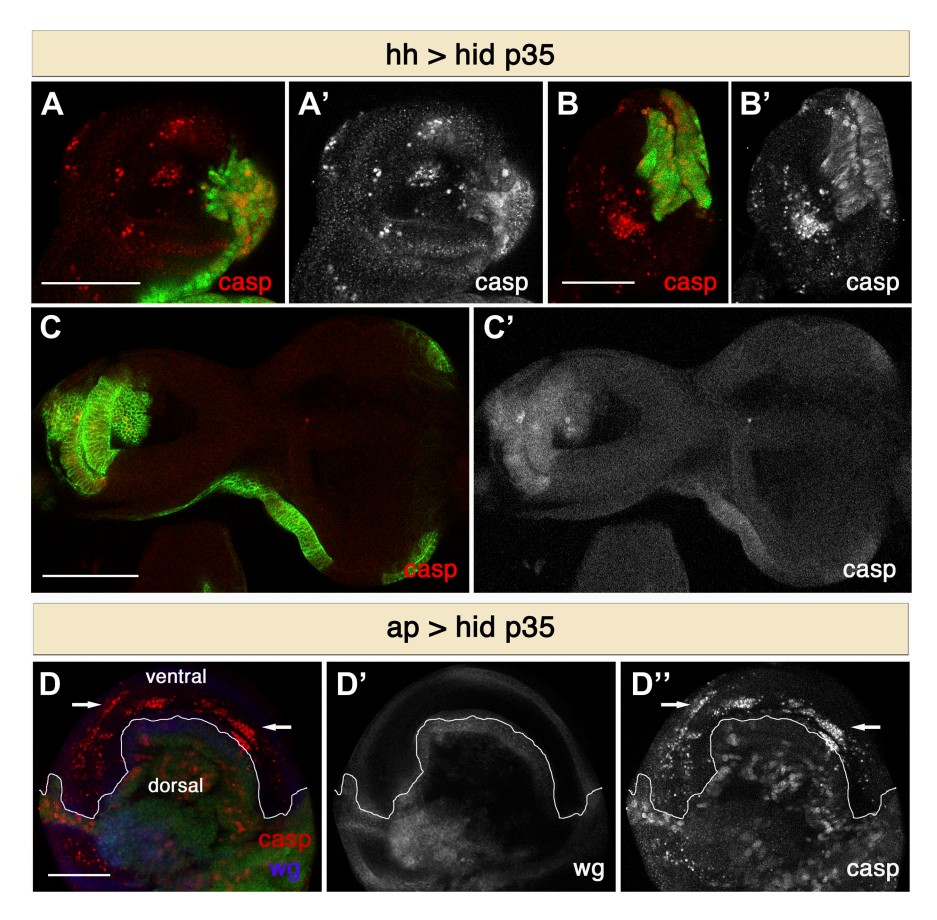

**Figure 2**. Non-autonomous apoptosis induced in different imaginal discs and with different drivers. (**A** and **B**) Haltere (**A**) and leg (**B**) discs of the genotype *hh-Gal4>UAS-GFP UAS-hid UAS-p35* also show non-autonomous apoptosis in the anterior compartment. (**C**) Eye-antenal discs of the genotype *hh-Gal4>UAS-myr-mRFP UAS-hid UAS-p35* (labeling with *UAS-myr-mRFP* is shown in green). In this case, non-autonomous apoptosis is not observed. (**D**) Wing disc of the genotype *ap-Gal4>UAS-GFP UAS-hid UAS-p35*. Undead cells in the dorsal compartment can be visualized by diffuse cleaved-caspase-3 staining (red in **D**, white in **D''**). Non-autonomous apoptosis was observed in the ventral compartment (arrows). Scale bars: 100 μm.

morphogenetic apoptosis (*Adachi-Yamada and O'Connor, 2002*). To examine this notion, we made experiments downregulating Wg or Dpp signals in the posterior compartment (*Figure 4B,C*). Under these conditions we failed to rescue non-autonomous apoptosis, which demonstrates that it is not a consequence of morphogenetic apoptosis or depends on the levels of Wg or Dpp (*Figure 4B,C*).

We also generated undead cells in the posterior compartment for a shorter period of time (3–4 days) to address the possibility that changes in growth, development and patterning could result in non-autonomous apoptosis. For this purpose, we employed the Gal4/Gal80^TS system to temporally control transgene expression (*McGuire et al., 2003*). Under these conditions we were able to generate discs that have a normal pattern, size and proliferation rates in the posterior compartment. Nevertheless, we again observed the induction of ectopic apoptosis in the anterior compartment (*Figure 4D*).

Taken together, these results indicate that apoptosis-induced non-autonomous apoptosis is not the consequence of cell competition, abnormal development, morphogenetic apoptosis or size compensation to prune excessive growth.

Since the previous experiments utilized undead cells, we investigated whether 'genuine' apoptotic cells have the same signaling capacity. For this purpose, we induced apoptosis in the posterior compartment by expressing *rpr* or *hid* alone, without *p35*. To allow for larval viability, we again used the conditional Gal4/Gal80^TS system to express pro-apoptotic proteins for only 48–72 hr. Under these conditions,

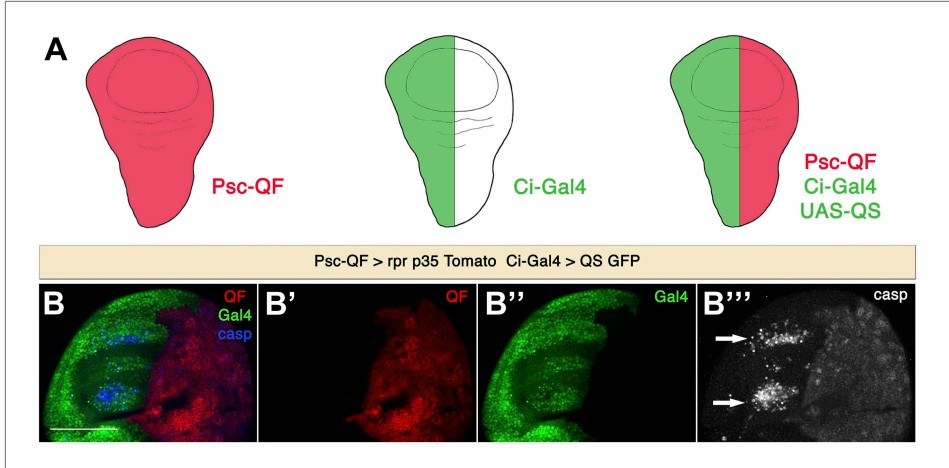

**Figure 3**. Apoptosis in the anterior compartment is non-autonomous. (**A**) Combination of Q and Gal4 systems to independently control expression in the anterior and posterior compartments. We make use of the *Psc-QF* driver, which is expressed ubiquitously in the wing disc (red), and the *Ci-Gal4* driver, which is expressed in the anterior compartment (green). By using *Ci-Gal4* to drive expression of the QS suppressor (*UAS-QS*), QF expression can be restricted to the posterior compartment. In this way, we can control transgene expression independently in the anterior compartment (with the Gal4 system, green) and in the posterior compartment (with the Q system, red). (**B**) Wing discs of the genotype *Psc-QF>QUAS-Tomato QUAS-rpr QUAS-p35 Ci-Gal4>UAS-GFP UAS-QS*. QF expression is restricted to the posterior compartment (red in **B** and **B'**), while Gal4 expression can be visualized in the anterior compartment (green in **B** and **B''**). Expression of *rpr* and *p35* in the posterior compartment using the Q system leads to generation of undead cells in the posterior compartment and the induction of non-autonomous apoptosis in the anterior compartment, as shown by cleaved caspase-3 staining (blue in **B**, white in **B'''**). Dying cells in the anterior compartment (arrows) are of anterior origin, as shown by the expression of *Ci-Gal4>UAS-GFP*. Scale bar: 100 μm.

large numbers of apoptotic cells were generated in the posterior compartment, and again we saw the induction of apoptosis in the anterior compartment (*Figure 5*). In this situation, the amount of non-autonomous apoptosis was generally less prominent compared to the use of undead cells, but in some cases it reached very high levels (*Figure 5C*). Hence, we conclude that genuine apoptotic cells have the capacity to induce non-autonomous apoptosis at a distance.

## JNK plays a role in the induction of non-autonomous apoptosis

The preceding results suggest the existence of a signal(s) emanating from apoptotic cells that can act at a distance to induce apoptosis across compartment borders, and we term this phenomenon 'Apoptosis-induced-Apoptosis (AiA)'. To gain insight into the underlying molecular mechanism, we considered a role of the JNK pathway since it is activated during stress-induced apoptosis in *Drosophila* and plays a role for the induction of Dpp and Wg in apoptotic cells (*Ryoo et al., 2004*; *Perez-Garijo et al., 2009*). For this purpose, we first examined the pattern of JNK activation by generating undead cells in the posterior compartment and used a *puckered-lacZ* construct to monitor JNK activity. *puckered* (*puc*), the sole *Drosophila* JNK-specific MAPK phosphatase, is a feedback antagonist of the JNK pathway (*Martin-Blanco et al., 1998*). Hence, the *puc-lacZ* line, which is an insertion in the *puckered* gene (*puc$^{E69}$*), serves both as a mutant for *puckered* and as a read-out for JNK activity. As expected, we saw very high *puc-lacZ* staining in undead cells, where the apoptotic loop keeps JNK constantly activated (*Shlevkov and Morata, 2012*). However, we also observed modest activation of JNK in dying cells in the anterior compartment (*Figure 6A*). Sometimes we also saw a trail of puc-lacZ activity between undead cells and dying cells. We also investigated the effect of increasing the activity of the JNK pathway using the above-mentioned mutant background for *puckered* (*puc$^{E69}$*). Normally, the generation of undead cells in the posterior compartment for 72 hr causes only a modest induction of apoptosis in the anterior compartment (*Figure 6B*). However, in a *puc$^{+/-}$* background, the amount of apoptosis in the anterior compartment was dramatically increased (*Figure 6C*). This suggests that JNK is involved in the induction of non-autonomous apoptosis.

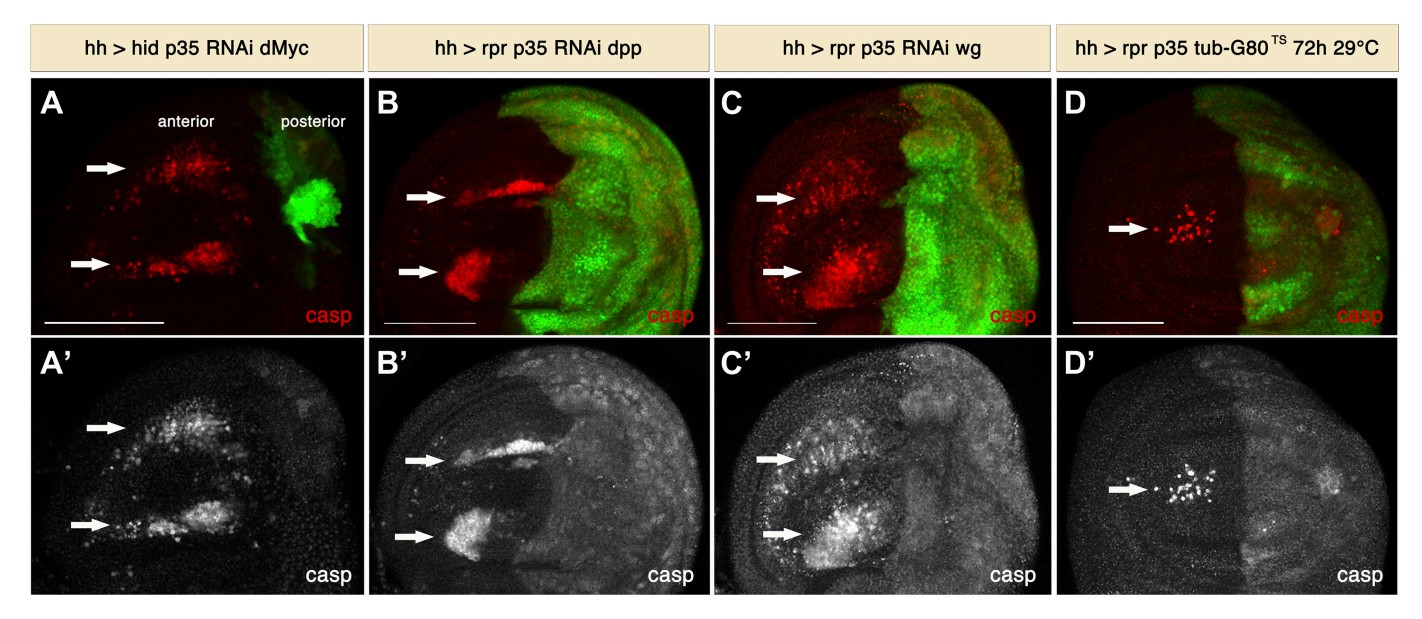

**Figure 4**. Signaling from apoptotic cells induces non-autonomous apoptosis. (**A**) Wing disc of the genotype *hh-Gal4>UAS-GFP UAS-hid UAS-p35 UAS-RNAi dMyc*. The posterior compartment is very reduced in size, but non-autonomous apoptosis is still present in the anterior compartment (arrows), as shown by staining with cleaved caspase-3 (red in **A**, white in **A'**). (**B** and **C**) Wing discs of the genotype *hh-Gal4>UAS-GFP UAS-rpr UAS-p35 UAS-RNAi-dpp* (**B**) and *hh-Gal4>UAS-GFP UAS-rpr UAS-p35 UAS-RNAi-wg* (**C**). Downregulation of the mitogenic signals Wg and Dpp produced by undead cells does not affect the amount of non-autonomous apoptosis, which is labeled with cleaved caspase-3 antibody (red in **B** and **C**; white in **B'** and **C'**). (**D**) Wing disc of the genotype *hh-Gal4>UAS-GFP UAS-rpr UAS-p35 tub-Gal80*<sup>TS</sup> grown at 29°C for the last 72 hr of larval development. Some apoptosis was still observed in the anterior compartment (arrows). Scale bars: 100 μm.

To further investigate the role of JNK in AiA, we downregulated the activity of this pathway using a mutant for *hemipterous* (*hep*), the *Drosophila* JNK-Kinase (dJNKK). We used hh-Gal4>UAS-hid UAS-p35 to generate undead cells in the posterior compartment and extensive induction of cell death in the anterior compartment (*Figure 6D*). Strikingly, apoptosis in the anterior compartment was completely blocked in male larvae hemizygous for *hep¹* (*Figure 6E*). Importantly, *hep¹* hemizygous mutants retained diffuse staining with caspase-3 in the posterior compartment, demonstrating that the generation of undead cells is not suppressed. Collectively, these experiments indicate that the JNK pathway plays an important role in AiA.

## JNK activation is required in cells induced to die in the anterior compartment

The previous experiments utilized mutant backgrounds for *puc* and *hep*. Under those conditions, both the posterior compartment (where undead cells are generated) and the anterior compartment (where non-autonomous apoptosis takes place) are mutant for those genes. The caveat of this approach is that reducing JNK activity in undead cells could potentially diminish their signaling capability. To gain insight into the role of JNK in AiA, we decided to specifically block JNK signaling in the anterior compartment, combining once more the Q and Gal4 systems. As previously shown, generation of undead cells in the posterior compartment by expression of *rpr* and *p35* using the Q system leads to non-autonomous apoptosis (*Figure 6F*). The advantage of this system is that Ci-Gal4 expression in the anterior compartment not only allows us to label the anterior compartment, but also to direct expression of any gene of interest specifically in the anterior compartment using the Gal4 system.

To block JNK activity in the anterior compartment, we drove expression of the RNAi of *basket* (*bsk*), the *Drosophila* JNK. Under these conditions, non-autonomous apoptosis is completely abrogated (*Figure 6G*). Since here we are not affecting JNK activity in undead cells, we can conclude that JNK is required for cells to die in the anterior compartment as a consequence of AiA.

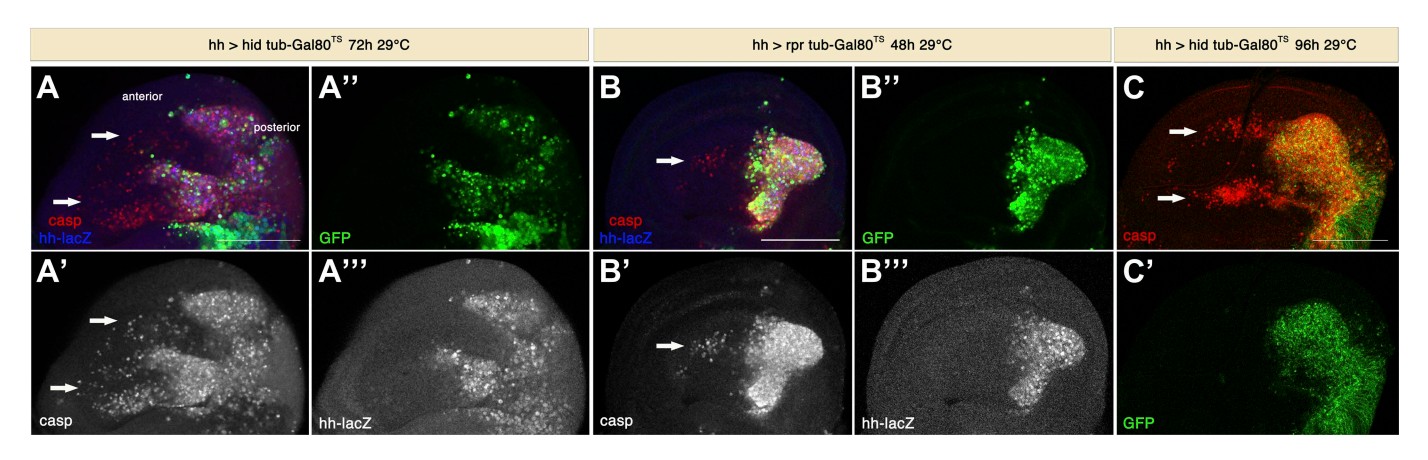

**Figure 5**. Genuine apoptosis also induces non-autonomous apoptosis. (**A**) Wing disc of the genotype *hh-Gal4>UAS-GFP UAS-hid tub-Gal80 TS* grown at 29°C for 72 hr. (**B**) Wing disc of the genotype *hh-Gal4>UAS-GFP UAS-rpr tub-Gal80TS* grown at 29°C for 48 hr. (**C**) Wing disc of the genotype *hh-Gal4>UAS-GFP UAS-hid tub-Gal80TS* grown at 29°C for 96 hr. In all cases, the posterior compartment is labeled by GFP expression (green). In **A** and **B**, the posterior compartment is also labeled with the *hh-lacZ* construct (blue in **A** and **B**, white in **A'''** and **B'''**). Genuine apoptosis can be visualized in the posterior compartment by punctate staining with cleaved caspase-3 (red in **A**–**C**, white in **A'** and **B'**). Non-autonomous apoptosis was also induced in the anterior compartment (arrows). Scale bars: 100 µm.

### Eiger produced by apoptotic cells is responsible for apoptosis-induced-apoptosis

One way by which the JNK pathway is activated in *Drosophila* is through the TNF ligand Eiger. Significantly, over-expression of Eiger can induce cell death through activation of the JNK pathway (*Igaki et al., 2002*; *Moreno et al., 2002*; *Kauppila et al., 2003*). The molecular basis of TNF-induced cell death in *Drosophila* has been well studied (*Kanda et al., 2002*; *Geuking et al., 2005*, *2009*; *Xue et al., 2007*; *Narasimamurthy et al., 2009*; *Kanda et al., 2011*; *Ma et al., 2012*). However, in vivo roles for Eiger-induced cell death have been elusive (*Igaki et al., 2009*; *Maezawa et al., 2009*; *Keller et al., 2011*). To investigate a possible role of Eiger in AiA, we first examined whether its expression is induced in undead cells (*Figure 7A*). Indeed, we saw significant up-regulation of Eiger in undead cells, consistent with the idea that Eiger may be produced by dying cells as a signal to induce non-autonomous apoptosis (*Figure 7A''*).

To critically test this hypothesis, we generated undead cells in the posterior compartment of wing discs homozygous mutant for *eiger*. Strikingly, the elimination of *eiger* function completely abrogated the non-autonomous apoptosis in the anterior compartment (*Figure 7B–E*). Importantly, under these conditions we still observed overgrown posterior compartments and a large amount of ectopic Wg signaling, as well as diffuse caspase staining, in undead cells (*Figure 7E*). This indicates that loss of Eiger does not impair JNK activation and mitogen production within undead cells.

In the previous experiments we used a condition where the entire disc is mutant for Eiger. We next investigated whether Eiger produced by undead cells is required for the induction of non-autonomous apoptosis. To address this question, we specifically downregulated Eiger levels in the posterior compartment with Eiger RNAi. Consistent with our model, we observed that AiA was significantly decreased by inhibiting Eiger specifically in undead cells (*Figure 8A–C*). Taken together, these experiments demonstrate that Eiger is a key signal that is generated by apoptotic cells to induce apoptosis of other cells at a distance (*Figure 8D*).

### In mice, cells undergoing apoptosis express TNF-α, which is crucial for hair cycle progression and the propagation of apoptosis

We next sought out to examine whether AiA occurs under a normal physiological setting and whether it can be expanded to mice. The mouse hair follicle (HF) is an ideal system for investigating AiA since cohort cell death naturally occurs at precise times in the HF cycle (*Lindner et al., 1997*; *Botchkareva et al., 2006*; *Tong and Coulombe, 2006*). The HF cycles between phases of growth (anagen), destruction

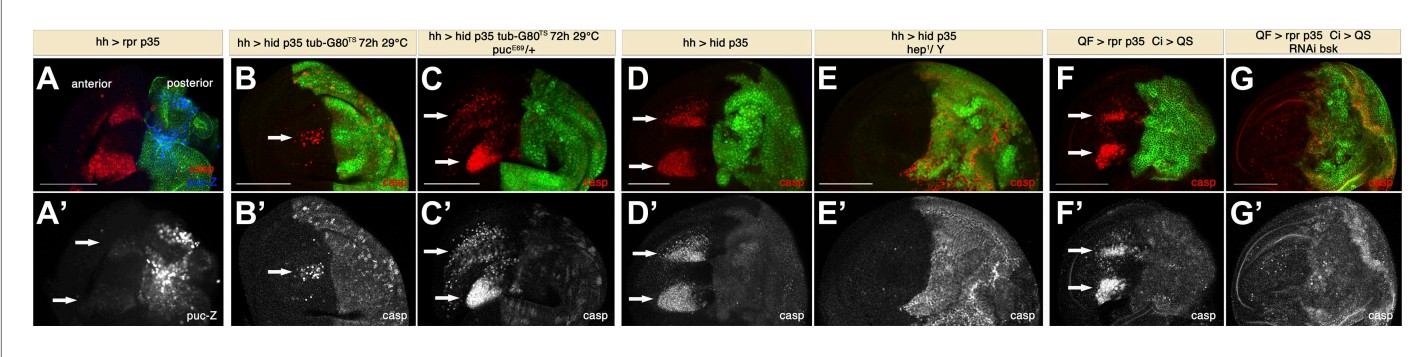

**Figure 6**. Apoptosis-induced apoptosis depends on JNK signaling. (**A**) Wing disc of the genotype *hh-Gal4>UAS-myr-mRFP UAS-rpr UAS-p35 puc^{E69}-lacZ/+* grown at 25°C (labeling of the posterior compartment with *UAS-myr-mRFP* is shown in green). Non-autonomous apoptosis in the anterior compartment is visualized by staining with cleaved caspase-3 antibody (red). *puc-lacZ* expression (blue in **A**, white in **A'**) reveals very strong activation of JNK pathway in undead cells and modest activation in dying cells in the anterior compartment (arrows). (**B**) Wing disc of the genotype *hh-Gal4>UAS-GFP UAS-hid UAS-p35 tub-Gal80^{TS}* grown at 29°C for the last 72 hr of larval development. A reduced amount of non-autonomous apoptosis was observed in the anterior compartment (arrows), as shown by staining with cleaved caspase-3 antibody (red in **B**, white in **B'**). (**C**) Wing disc of the genotype *hh-Gal4>UAS-GFP UAS-hid UAS-p35 tub-Gal80^{TS} puc^{E69}-lacZ/+* grown at 29°C during the same period of time. The amount of apoptosis in the anterior compartment was greatly increased (arrows). (**D**) Wing disc of the genotype *hh-Gal4>UAS-myr-mRFP UAS-hid UAS-p35* grown at 25°C (labeling of the posterior compartment with *UAS-myr-mRFP* is shown in green). Cleaved caspase-3 staining (red in **D**, white in **D'**) reveals a large amount of non-autonomous apoptosis in the anterior compartment (arrows). (**E**) Downregulation of the JNK pathway suppresses non-autonomous apoptosis, even though undead cells are still present in the posterior compartment, as shown by caspase-3 staining (red in **E**, white in **E'**). Genotype: *hh-Gal4>UAS-myr-mRFP UAS-hid UAS-p35 hep^1/Y* (mRFP is also shown in green). (**F**) Wing disc of the genotype *Psc-QF>QUAS-Tomato QUAS-rpr QUAS-p35 Ci-Gal4>UAS-QS*. Expression of the driver QF is restricted to the posterior compartment (labeling of the compartment with *QUAS-Tomato* is shown in green). Cleaved caspase-3 (red in **F**, white in **F'**) staining labels undead cells in the posterior compartment and apoptotic cells in the anterior compartment (arrows). (**G**) Inhibition of the JNK pathway specifically in the anterior compartment completely suppresses non-autonomous apoptosis. Genotype: *Psc-QF>QUAS-Tomato QUAS-rpr QUAS-p35 Ci-Gal4>UAS-QS UAS-RNAi-bsk*. Caspase-3 staining labels intact undead cells in the posterior compartment (red in **G**, white in **G'**). Labeling of the compartment with *QUAS-Tomato* is shown in green. Scale bars: 100 µm.

(catagen) and rest (telogen) (*Hardy, 1992*; *Fuchs, 2007*). During catagen, apoptosis leads to degeneration of the lower two-thirds of the HF while the upper part remains intact (*Lindner et al., 1997*; *Tong and Coulombe, 2006*) (*Figure 9A*). This portion encompasses the bulge, which houses the HF stem cells required for the generation of the new HF (*Rompolas et al., 2012*). The first catagen phase commences at postnatal day 16 (P16) and hence we isolated skin at this time point. At P16, we could easily detect HFs with a large number of apoptotic cells. These cells displayed apoptotic morphology such as membrane blebbing, condensed nuclei and were positive for cleaved capase-3 (*Figure 9B*). It was previously shown that the HF cycle is dependent upon TNF signaling (*Hoffmann et al., 1996*; *Ruckert et al., 2000*; *Tong and Coulombe, 2006*), but the cellular source of TNF-α has not been determined. If AiA was involved, TNF-α should be produced by apoptotic cells. We examined TNF-α expression and found that apoptotic cells in the HF expressed high levels of TNF-α (*Figure 9B*). Significantly, TNF-α was not detected at any other location besides apoptotic cells. These results are consistent with the idea that apoptotic cells are the source of pro-death signals during catagen and that AiA contributes to the destruction of the HF.

We went on to inhibit the function of TNF-α in vivo by injecting mice with a TNF-α neutralizing antibody (mp6-xt22) (*Abrams et al., 1992*). Injections were commenced on P14 littermates, when HFs were in anagen, and a non-specific IgG was used as control. Antibodies were injected daily from P14-P17 and, as expected, at P17.5 control dorsal skin HFs displayed a morphology typical for late catagen (*Figure 9C*). In contrast, upon inhibition of TNF-α HFs escaped destruction and had a morphology reminiscent of anagen or very early catagen (*Figure 9D*). We next extended this analysis to tail HFs. We analyzed tailskin whole mounts and observed again an inhibition in the destruction of the HF and also a clear loss of synchronicity (*Figure 9E–G*). To assess effects on apoptosis, we used an antibody that specifically detects the cleaved form of caspase-3. Compared to controls, TNF-α inhibited animals had dramatically reduced numbers of cleaved caspase-3-positive cells, indicating that TNF-α promotes

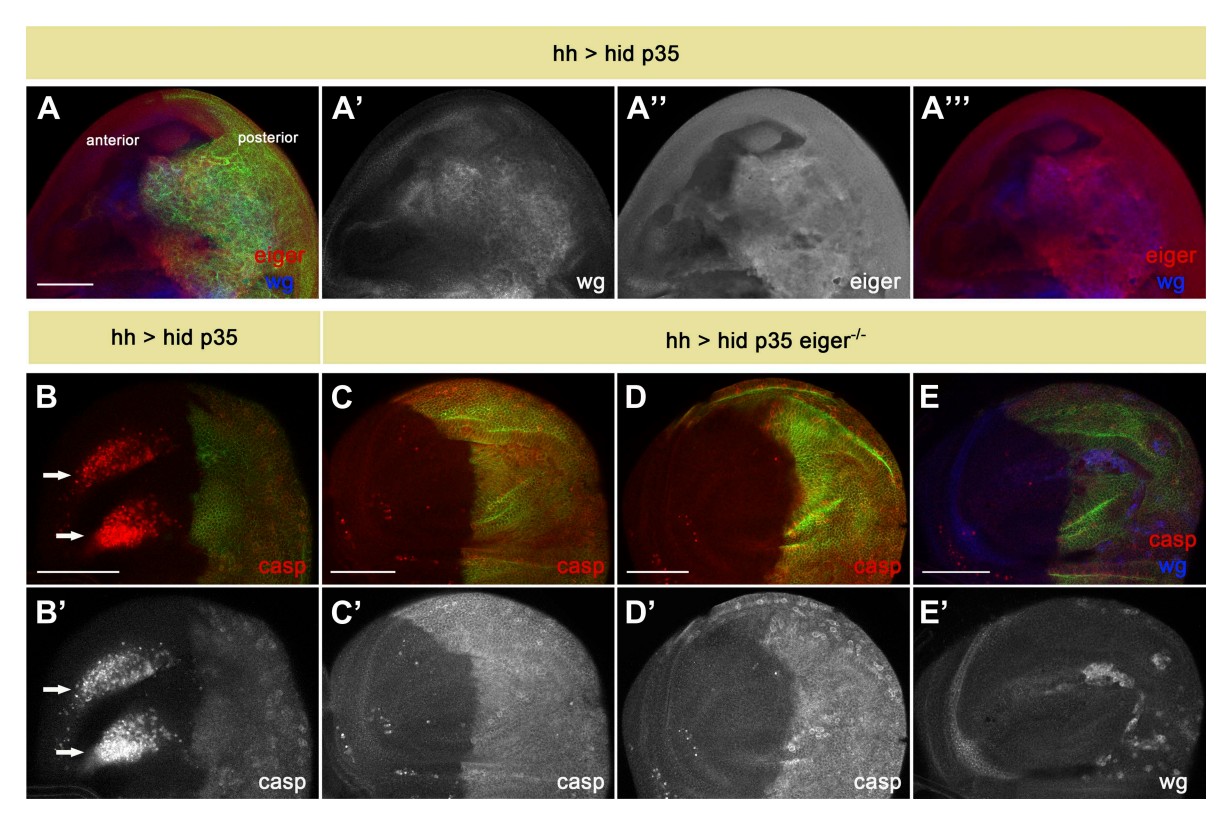

**Figure 7**. Eiger is responsible for apoptosis-induced apoptosis. (**A**) Wing disc of the genotype *hh-Gal4>UAS-myr-mRFP UAS-hid UAS-p35*. The posterior compartment shows ectopic *wg* expression (blue in **A** and **A'''**, white in **A'**). Eiger levels are also elevated in undead cells (red in **A** and **A'''**, white in **A''**). (**B**) Wing disc of the same genotype stained with anti-cleaved caspase-3 antibody (red in **B**, white in **B'**). Non-autonomous apoptosis in the anterior compartment is indicated by arrows. (**C–E**) Apoptosis-induced apoptosis is suppressed in an *eiger* mutant background, as shown by cleaved caspase-3 staining (red in **C–E**, white in **C'** and **D'**). Ectopic expression of Wg is observed in the posterior compartment (blue in **E**, white in **E'**), suggesting that undead cells are not compromised in their signaling capacity under these conditions. Genotypes: *hh-Gal4>UAS-myr-mRFP UAS-hid UAS-p35 eiger¹/eiger¹* (**C**), *hh-Gal4>UAS-myr-mRFP UAS-hid UAS-p35 eiger¹/eiger³* (**D**) and *hh-Gal4>UAS-myr-mRFP UAS-hid UAS-p35 eiger³/eiger³* (**E**). The posterior compartment is labeled with *UAS-myr-mRFP* (green) in all cases. Scale bars: 100 μm.

apoptosis in this system (*Figure 9E,F*). Taken together, these data suggest that in mice TNF-α is expressed by apoptotic cells in the skin, and that it plays a physiological role to orchestrate cohort cell death during HF regression.

## Discussion

It is becoming clear that apoptosis is not a passive phenomenon where dying cells merely die and are silently removed from the tissues. Instead, apoptotic cells have the capacity to produce proliferative signals, such as Wg and Dpp, thus serving as a crucial driving force in wound healing, regeneration and tumor formation in a variety of different organisms (*Huh et al., 2004*; *Perez-Garijo et al., 2004*; *Ryoo et al., 2004*; *Chera et al., 2009*; *Li et al., 2010*; *Pellettieri et al., 2010*; *Huang et al., 2011*). Here, we demonstrate that signaling by apoptotic cells is not limited to the production of proliferative signals: dying cells can generate pro-apoptotic signals that induce apoptosis in neighboring cells. We show that the induction of this apoptosis is triggered by Eiger, the ortholog of TNF in *Drosophila*, which in turn activates the JNK pathway and leads to cell death in a non-autonomous manner. Furthermore, we were able to provide evidence for the existence of AiA under physiological conditions in mice. We found that during catagen, the regressive phase of the hair cycle, apoptotic cells in the lower and transient portion of the HF also express TNF-α. Significantly, TNF-α is required for coordinated cell death in the HF. Taken together, these results suggest that AiA plays an important physiological role for the coordination of cohort cell death.

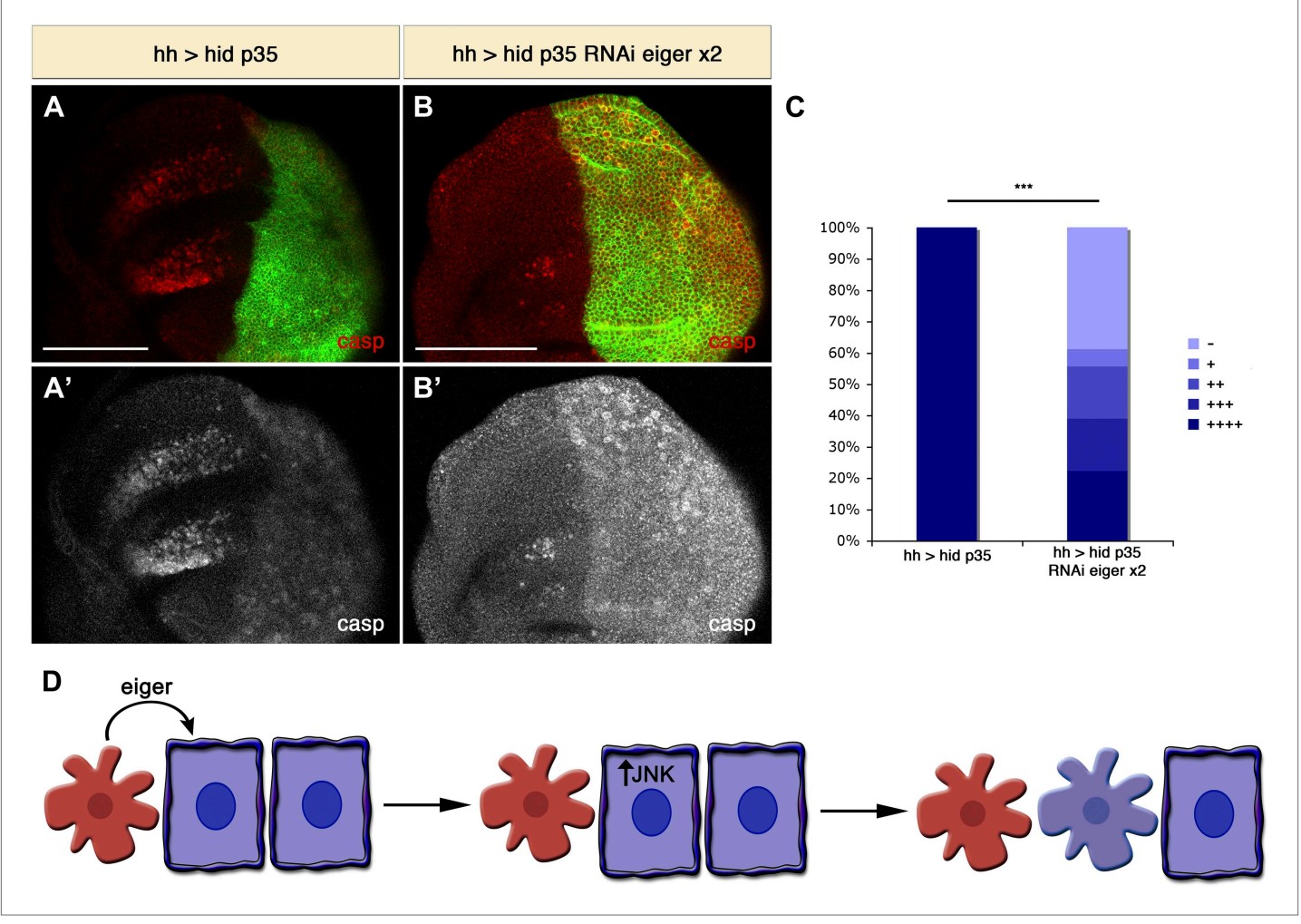

**Figure 8**. Downregulation of Eiger in undead cells significantly reduces apoptosis-induced apoptosis. (**A**) Wing disc of the genotype *hh-Gal4>UAS-myr-mRFP UAS-hid UAS-p35* grown at 29°C. (**B**) Wing disc of the genotype *hh-Gal4>UAS-myr-mRFP UAS-hid UAS-p35 UAS-RNAi-eiger KK108814 UAS-RNAi-eiger KK108814*, also grown at 29°C. Apoptosis is shown in all cases by staining with cleaved caspase-3 antibody (red in **A** and **B**, white in **A'** and **B'**). The posterior compartment is labeled with *UAS-myr-mRFP* (green). (**C**) Measurement of the levels of apoptosis-induced apoptosis in both conditions. The amount of non-autonomous apoptosis is graded in five different categories, from widespread apoptosis (++++) to no apoptosis (−). Downregulation of Eiger in the posterior compartment significantly decreases AiA. p<0.001. (**D**) Model for apoptosis-induced apoptosis. Apoptotic cells produce Eiger, the *Drosophila* TNF homolog, which activates the JNK pathway in neighboring cells, leading to cell death in a non-autonomous manner. Scale bars: 100 µm.

## Apoptosis-induced-apoptosis as a mechanism to propagate cell death within a tissue

Our experiments demonstrate that induction of apoptosis in one compartment results in induction of non-autonomous apoptosis in the neighboring compartment. This is true under many different conditions: both when we generate undead cells (expressing *rpr/hid* and *p35*) or upon induction of genuine apoptosis (expressing *rpr/hid* alone); once there is ectopic expression of mitogens that leads to excessive proliferation and growth or while blocking mitogenic production or growth of the compartment.

One intriguing observation is that this non-autonomous cell death usually displays a pattern consisting of two groups of cells in the wing pouch. One possible explanation for this is that the affected cells are the most susceptible to the death signal. In fact, the regions of the wing pouch where we observe the non-autonomous cell death are also more prone to cell death as a response to different apoptotic stimuli, such as irradiation or *hid* over-expression (***Milan et al., 1997***; ***Moon et al., 2005***).

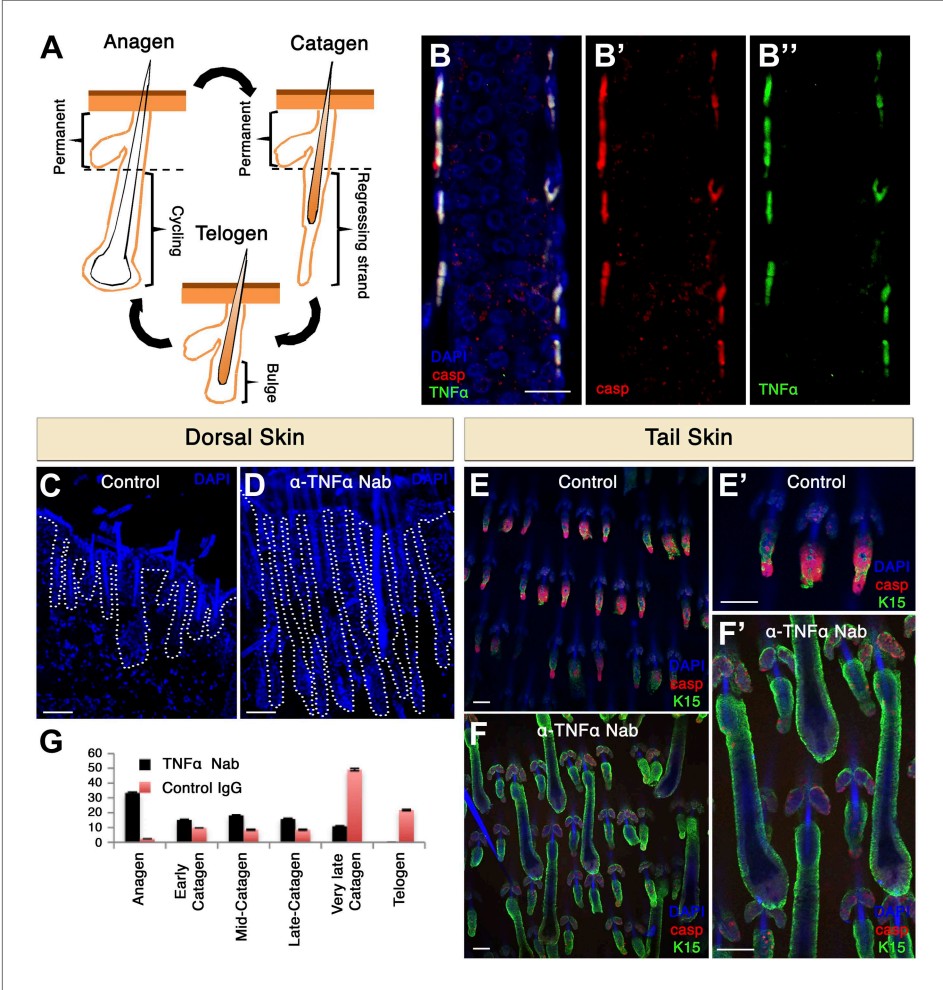

**Figure 9**. TNF-α is expressed in apoptotic HF cells during catagen and is essential for coordinated apoptosis and hair cycle progression in mice. (**A**) Schematic diagram depicting the HF cycle. HFs cycle between phases of growth (anagen), destruction (catagen) and rest (telogen). During catagen, apoptosis leads to the degeneration of the lower two-thirds of the HF. Post-catagen, the HF enters the quiescent telogen phase, which later on enters a new cycle of hair growth (anagen). (**B**) Immunofluorescence staining indicating that apoptotic HF cells express TNF-α during catagen (P16) in wild-type conditions. Apoptotic cells are labeled with cleaved caspase-3 (red) and TNF-α staining is shown in green. (**C–F**) Neutralizing TNF-α in vivo impairs the HF cycle progression as a result of decreased apoptosis. (**C** and **D**) DAPI staining of dorsal skin demonstrates impaired HF cycle in response to TNF-α neutralization (P17.5). (**E** and **F**) Immunofluorescence staining of tail whole mounts indicates decreased cleaved caspase-3 staining (red) and desynchronization of the HF cycle in TNF-α neutralized mice (P17.5). (**G**) Statistical analysis of the HF phase in control and TNF-α inhibited mice. Scale bars: 20 µm in **B**, 100 µm in **C–F**.

Another possibility to explain why we observe cell death at a distance would be that dying cells are producing other signals that inhibit apoptosis. This protective signal would diffuse only short range, and in this way the distance of cells to the border would determine the ratio between the pro-apoptotic and the protective signal, tipping the balance in favor of death or survival. In fact, it has been shown that cells neighboring apoptotic cells downregulate Hippo pathway and consequently activate Diap1 (*Grusche et al., 2011*; *Sun and Irvine, 2011*). Another good candidate for such an anti-apoptotic signal would be Wg, as it is expressed in an opposite pattern from the non-autonomous apoptosis and is also diffusing from apoptotic cells in the posterior compartment. However, we tried to modify Wg levels in different ways and we did not observe any change in the apoptosis pattern (data not shown).

On the other hand, in physiological conditions such as the coordinated cell death of HF cells observed in mice, it would be expected that signaling between apoptotic cells would occur at a much

shorter range, probably affecting the immediate neighbors. In any case, the observation that TNF-α is exclusively detected in apoptotic cells and the fact that its inhibition leads to desynchronization of the HF cycle strongly suggests that AiA can be a mechanism to coordinate cell death within a tissue.

## The TNF and JNK pathways are required for apoptosis-induced-apoptosis

In our experimental systems, AiA requires both the TNF and JNK signaling pathways. Eiger is produced by apoptotic cells in the posterior compartment of the wing disc and it activates JNK in cells of the neighboring compartment, inducing them to die. Downregulation of Eiger in the posterior compartment or JNK in the anterior compartment was able to suppress AiA. However, it remains to be elucidated whether Eiger directly diffuses to the cells in the anterior compartment, or if some other mechanism is responsible for the activation of JNK in dying cells in the anterior compartment. Recently, it was shown that, upon wounding, JNK activity can be propagated at a distance through a feed-forward loop (*Wu et al., 2010*). Significantly, AiA is not restricted to the *Drosophila* wing disc. We obtained evidence for a role of TNF-α-mediated AiA during the destruction of the hair follicle (HF) in catagen, the regressive phase of the hair cycle (*Figure 9*). TNF-α plays a known role to promote cell death and has been previously implicated in HF progression, wound healing and regeneration (*Werner and Grose, 2003*; *Tong and Coulombe, 2006*; *Bohm et al., 2010*). However, the cellular source of TNF-α remained unknown and it was previously not appreciated that apoptotic cells can be the source of these signals. Our results suggest that AiA and at least some of the underlying mechanism have been conserved in evolution to promote coordinated cell death.

## Significance of AiA in development and disease

The observation that apoptotic cells can signal to other cells in their environment and instruct them to die has potentially many important implications. On the one hand, there are situations where propagation of an apoptotic stimulus may be a useful mechanism to achieve the rapid and coordinated death of large cell populations. Our experiments in mice show that this can be the case during the catagen phase of the HF cycle. There are many other examples of cell death being used during development to sculpt tissues and organs, including the removal of structures during metamorphosis (tadpole tail, larval organs in insects, elimination of inappropriate sex organs in mammals, deletion of the amnio serosa during insect embryogenesis) and the separation of digits through apoptosis of the interdigital webbing in many vertebrates (*Glucksmann, 1951*; *Jacobson et al., 1997*; *Fuchs and Steller, 2011*). In all these cases, AiA may facilitate cohort behavior and contribute to the rapid and complete elimination of large fields of cells.

Propagation of cell death may also be an efficient way to prevent infection. It is known that cells respond to viral infection by entering apoptosis and in this way impede the replication of the virus (*Barber, 2001*). The process of AiA would extend apoptosis to the neighboring cells, preventing also their infection and thus avoiding the spread of the virus.

However, propagation of apoptosis may be detrimental in pathological conditions where excessive cell death underlies the etiology of the disease. This may be the case for neurodegenerative disorders, hepatic diseases, cardiac infarction, etc (*Thompson, 1995*; *Favaloro et al., 2012*). In all these cases it remains to be studied whether extensive amounts of apoptosis that are observed in the affected tissues are a direct consequence of cell damage in an autonomous manner or if part of the cell loss could be attributed to a process of propagation through AiA.

Finally, AiA may play a role in cancer. It is known that radiotherapy in humans can induce biological effects in non-irradiated cells at a considerable distance, a phenomenon called radiation-induced bystander effect (*Hei et al., 2011*; *Prise and O'Sullivan, 2009*). Our current findings provide a possible explanation for some of these effects. Therefore, large-scale induction of apoptosis by AiA may contribute to successful cancer therapy. TNF family proteins are being used as models for drug development aimed to treat cancer (*Ashkenazi, 2008*). Furthermore, Eiger, the only TNF member in *Drosophila*, has a known role in the elimination of pre-tumoral *scrib⁻* clones (*Igaki et al., 2009*; *Ohsawa et al., 2011*). In addition, cell competition induces cell death even in aggressive *scrib⁻Ras^{V12}* tumors, raising the possibility that AiA is induced during tumor initiation, which may affect the tumor microenvironment and ultimately tumor growth (*Menendez et al., 2010*). It is well known that TNF can play both tumor-promoting and tumor-suppressing roles, but AiA has not been investigated in this context (*Pikarsky and Ben-Neriah, 2006*; *Vainer et al., 2008*). Future studies will shed new light on the relevance of signaling by apoptotic cells and the implications of this signaling mechanism in different scenarios.

## Material and methods

### Fly strains and crosses

All flies were raised in standard fly food at 25°C unless indicated otherwise.

To generate undead cells, UAS-rpr UAS-p35 or UAS-hid UAS-p35 flies were crossed to the appropriate drivers (hh-Gal4, ap-Gal4 and Ci-Gal4 were a gift from G Morata). UAS-GFP and UAS-myr-mRFP (Bloomington Stock Center) were used to visualize the compartments. As a control for normal apoptosis we used the hh-Gal4 UAS-GFP line.

To generate posterior compartments of reduced size we used the UAS-RNAi dMyc (P[TRiP. JF01761]attP2) stock #25783 from TRiP (available at Bloomington Stock Center). To downregulate Wg and Dpp in the posterior compartment we used the UAS-RNAi wg GD#13352 (VDRC) and the UAS-RNAi dpp (P[TRiP.JF01371]attP2) stock #25782 (TRiP, available at Bloomington Stock Center), respectively.

For temporal transgene expression we used the Gal4/Gal80 $^{TS}$ system (**McGuire et al., 2003**). We crossed tub-Gal80 TS; hh-Gal4 flies to UAS-hid; UAS-p35 lines (for transient generation of undead cells) or UAS-hid or UAS-rpr lines (for induction of genuine apoptosis). Flies were raised at 18°C, where the driver Gal4 is inhibited by the tub-Gal80, and the larvae were shifted at 29°C at different time points to initiate Gal4 activity. Late third instar larvae were dissected after different periods of time at 29°C, as mentioned in the text.

We used a hh-lacZ line to label the posterior compartment (a gift from G Morata).

To modify JNK activity we used puc$^{E69}$ and hep$^1$ mutants (**Martin-Blanco et al., 1998**). To eliminate Eiger function we used eiger$^1$ and eiger$^3$ mutants (a gift from N Baker) (**Igaki et al., 2002**). To downregulate Eiger specifically in the posterior compartment we used UAS-RNAi eiger KK#108814 from VDRC. To downregulate JNK specifically in the anterior compartment we used the UAS-RNAi bsk GD#34138 from VDRC.

For the experiments using the Q system we used the line QUAS-mtdTomato, Psc-QF and the UAS-QS (stocks #30043 and #30033, respectively, from Bloomington Stock Center).

### Construction of QUAS-rpr and QUAS-p35

The *reaper* and *p35* inserts were PCR amplified from genomic DNA from flies containing the *UAS-p35* and *UAS-rpr* transgenes using the primers ATAGAGGCGCTTCGTCTACGG and CCCATTCATCA GTTCCATAGGTTG.

PCR products were digested with EcoRI for cloning into pQUAST (Addgene plasmid 24349). The sequence of the construct was verified by DNA sequencing.

For PCR amplifications we used the PfuUltra DNA polymerase (Agilent Technologies, Inc., Santa Clara, CA).

DNA constructs were sent for injection to Bestgene Inc. and one transgenic line for each construct was selected on the third chromosome.

### Histochemistry

Imaginal discs were dissected, fixed and stained as described previously (**Perez-Garijo et al., 2004**) using the following antibodies: anti-cleaved caspase-3 (1:200, Cell Signaling Technologies, Danvers, MA), anti-Wg 4D4 (1:50, DSHB, Iowa City, IA), mouse anti-β-Gal (1:50, DSHB) and anti-Eiger (1:500, gift from M Miura). Secondary antibodies were used 1:200 and purchased from Jackson Laboratories. Discs were then mounted in Vectashield (Vector Laboratories, Burlingame, CA). For TUNEL stainings we used the ApopTag in situ Apoptosis Detection Kit (Chemicon, Millipore, Billerica, MA) and followed the instructions provided by the kit. Images were taken with a LSM710 (Zeiss) confocal microscope and subsequently processed using Adobe Photoshop.

Mice skin harvested at P16 and P17 was embedded in OCT. Cryosections were fixed in 4% PFA for 10 min and blocked with 10% NGS, 0.5% and 2% BSA for 2 hr and primary antibodies were placed in blocking solution overnight. The following day secondary antibodies were used and sections were mounted with Vectashield (Vector Laboratories). Images were taken with a LSM710 (Zeiss) confocal microscope and subsequently processed using Adobe Photoshop. Tail whole mounts were performed as in **Braun et al. (2003)**. Antibodies used: anti-cleaved caspase-3 (1:100, Cell Signaling Technologies), LEAF purified anti-mouse TNF-α (1:100, Biolegends, San Diego, CA), anti-K15 (1:1000, Abcam, Cambridge, United Kingdom).

## TNF-α neutralization in mice

For in vivo neutralization experiments C57B littermate mice were broken into two groups (n = 7). Each group received either a non-specific Rat LEAF purified IgGI (10 µg/gr, Biolegends) or LEAF purified anti-mouse TNF-α (10 µg/gr, Biolegends). Antibodies were injected daily from P14-P17 in 200 µl of sterile saline subcutaneously.

## Acknowledgements

We thank our colleagues in the Steller lab for helpful discussion; J Menendez, J Rodriguez, S Benjamin, C Sandu, X Lee and F Bejarano for critically reading of the manuscript; G Morata, N Baker, the TRiP at Harvard Medical School (NIH/NIGMS R01-GM084947), Vienna *Drosophila* RNAi Center (VDRC) and Bloomington Stock Center for fly stocks; Bestgene Inc. for fly injection; and M Miura for anti-Eiger antibody. APG was a fellow of the Fundacion Ramon Areces. HS is an Investigator of the Howard Hughes Medical Institute.

## Additional information

### Funding

| Funder | Grant reference number | Author |
| --- | --- | --- |
| Howard Hughes Medical Institute | | Hermann Steller |
| National Institutes of Health | R01-AR050452 | Hermann Steller |
| Fundacion Ramon Areces | | Ainhoa Pérez-Garijo |

The funders had no role in study design, data collection and interpretation, or the decision to submit the work for publication.

### Author contributions

AP-G, YF, Conception and design, Acquisition of data, Analysis and interpretation of data, Drafting or revising the article; HS, Conception and design, Drafting or revising the article

### Ethics

Animal experimentation: This study was performed in strict accordance with the recommendations in the Guide for the Care and Use of Laboratory Animals of the National Institutes of Health. All of the animals were handled according to approved Institutional Animal Care and Use Committee (IACUC) protocols (#11417) of the Rockefeller University.

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
