## [Decision Letter]

Thank you for sending your work entitled “Apoptosis-induced apoptosis: propagation of cell death through the TNF pathway” for consideration at *eLife*. Your article has been favorably evaluated by a Senior editor, a Reviewing editor, and three reviewers.

All three reviewers thought that the data presented in your manuscript clearly describe a new phenomenon, termed “Apoptosis-induced apoptosis”(AiA). AiA occurs when apoptotic cells trigger apoptosis of cells at a distance, which are otherwise not programmed to die. The data support the conclusion that this phenomenon occurs in *Drosophila* imaginal discs and that there may be a potentially similar mechanism occurring in the hair follicle in the mouse. Before we reach a final decision, please consider responding to the following points made by the reviewers:

1) Perform *puc-lacZ* or TRE-GFP labelings in hh>hid>p35 discs. While you show a requirement of the JNK pathway for AiA, it is important to see where exactly JNK is activated. Is it only activated in the two populations of cells that undergo AiA, or alternatively is it activated in a broader domain and only the cells in the dying region respond by undergoing apoptosis?

2) The non-autonomous cell death in the discs could theoretically be due to leaky expression of the Gal4 driver or from the UAS construct alone. This possibility should be discussed. Leaky driver expression is in part dealt with in the QS/Gal4 experiment. However, it is important that the *UAS-hid* and *UAS-rpr* controls be shown.

3) The TNF staining in the hair follicle shows a surprisingly strong overlap with caspase staining, which does support the apoptosis induced apoptosis model. However, the neutralizing antibody injection does not prove that the TNF produced in those cells promotes catagen. You should be more careful about how you state the conclusions for this work. You can’t say that TNF expression in apoptotic HF cells is “essential” for apoptosis, unless you knock out expression in only those cells.

---

## [Author Response]

*1) Perform* puc-lacZ *or TRE-GFP labelings in hh>hid>p35 discs. While you show a requirement of the JNK pathway for AiA, it is important to see where exactly JNK is activated. Is it only activated in the two populations of cells that undergo AiA, or alternatively is it activated in a broader domain and only the cells in the dying region respond by undergoing apoptosis*?

We completely agree with the reviewers that this is an important point. Therefore, we performed *puc-lacZ* staining experiments and included the results in a new panel in Figure 5. We observed the highest levels of puc-lacZ in undead cells, where the apoptotic loop keeps JNK constantly activated. However, we also saw modest activation of JNK in the dying cells in the anterior compartment, and in cases we detected a trail of *puc-lacZ* between undead cells and dying cells.

*2) The non-autonomous cell death in the discs could theoretically be due to leaky expression of the Gal4 driver or from the UAS construct alone. This possibility should be discussed. Leaky driver expression is in part dealt with in the QS/Gal4 experiment. However, it is important that the* UAS-hid *and* UAS-rpr *controls be shown*.

We agree that this is an important point and therefore thoroughly examined the possibility of leaky driver expression. Based on our analyses we are very confident that this is not the case. First, we addressed this issue using the QF/Gal4 system (Figure 2). Unfortunately, the same type of experiment cannot be performed for over-expression of *hid* or *rpr* alone. The Psc-QF>QUAS-*rpr/hid*; Ci-Gal4>UAS-QS flies are lethal, and it is not possible to temporally control the expression of rpr/hid since the QS inhibitor is used for spatial control. Nevertheless, evidence for non-leakiness under these conditions is provided in Figure 4. Here we use posterior compartment markers (GFP and *hh-lacZ*) to show that anterior dying cells clearly lack these marks.

*3) The TNF staining in the hair follicle shows a surprisingly strong overlap with caspase staining, which does support the apoptosis induced apoptosis model. However, the neutralizing antibody injection does not prove that the TNF produced in those cells promotes catagen. You should be more careful about how you state the conclusions for this work. You can't say that TNF expression in apoptotic HF cells is “essential” for apoptosis, unless you knock out expression in only those cells*.

We fully agree and we have revised the text accordingly. Even though TNF expression was exclusively detected in apoptotic cells, eliminating its expression only in those cells would be technically very difficult. We rephrased the relevant text and figure legend to avoid any over-statements.